# Amenamevir, a Helicase-Primase Inhibitor, for the Optimal Treatment of Herpes Zoster

**DOI:** 10.3390/v13081547

**Published:** 2021-08-05

**Authors:** Kimiyasu Shiraki, Shinichiro Yasumoto, Nozomu Toyama, Hiroaki Fukuda

**Affiliations:** 1Faculty of Nursing, Senri Kinran University, 5-25-1 Fujishirodai, Suita, Osaka 565-0873, Japan; 2Yasumoto Dermatology Clinic, Tsukushino, Fukuoka 818-0083, Japan; shinyhc55103@extra.ocn.ne.jp; 3Toyama Dermatologic Clinic, Aburatsu, Nichinan City, Miyazaki 887-0001, Japan; nontoyama@miyazaki.med.or.jp; 4Maruho Co., Ltd., Nakatsu, Osaka 531-0071, Japan; fukuda_cis@mii.maruho.co.jp

**Keywords:** amenamevir, helicase-primase inhibitor, herpes simplex virus, varicella-zoster virus, herpes zoster, antivirals

## Abstract

Acyclovir, valacyclovir, and famciclovir are used for the treatment of herpes simplex virus (HSV) and varicella-zoster virus (VZV) infections. Helicase-primase inhibitors (HPIs) inhibit replication fork progression that separates double DNA strands into two single strands during DNA synthesis. The HPIs amenamevir and pritelivir have novel mechanisms of anti-herpetic action, and their once-daily administration has clinical efficacy for genital herpes. Among HPIs, amenamevir has anti-VZV activity. The concentrations of HSV-1 and VZV required for the 50% plaque reduction of amenamevir were 0.036 and 0.047 μM, respectively. We characterized the features of amenamevir regarding its mechanism, resistance, and synergism with acyclovir. Its antiviral activity was not influenced by the viral replication cycle, in contrast to acyclovir. A clinical trial of amenamevir for herpes zoster demonstrated its non-inferiority to valacyclovir. To date, amenamevir has been successfully used in over 1,240,000 patients with herpes zoster in Japan. Post-marketing surveillance of amenamevir in Japan reported side effects with significant potential risk identified by the Japanese Risk Management Plan, including thrombocytopenia, gingival bleeding, and palpitations, although none of these were serious. The clinical efficacy and safety profiles of amenamevir were established in patients with herpes zoster. Therefore, amenamevir as an HPI opens a new era of anti-herpes therapy.

## 1. Introduction

Varicella-zoster virus (VZV) infection, which causes varicella and herpes zoster, is treated with antivirals. Elion developed acyclovir for systemic administration to treat herpes simplex virus (HSV) and VZV infections [1,2,3]. Penciclovir, valacyclovir, and famciclovir are currently used for preventing and treating HSV and VZV infections [4].

The antiherpetic drugs valacyclovir (acyclovir) and famciclovir (penciclovir) have been developed for the treatment and prevention of apparent HSV and VZV infections. New antiherpetic drugs with different mechanisms of action have been developed as novel helicase-primase (HP) inhibitors (HPIs) of HSV and VZV. Double-stranded DNA is separated into two single strands (replication fork) before DNA synthesis, and complementary strands are synthesized from each DNA strand to produce two new double-stranded DNA molecules during DNA replication, as shown in Figure 1. The HP complex unwinds viral DNA at the replication fork, separating double-stranded DNA into two single strands, and synthesizing RNA primers followed by Okazaki fragments in the lagging strand for DNA synthesis. Then, DNA polymerase initiates complementary DNA synthesis in the two separated DNA strands. The HP complex consists of three proteins: HSV UL5/VZVORF55 (helicase), HSV UL52/VZVORF6 (primase), and HSV UL8/VZVORF52 (cofactor). 

A 2-amino thiazole compound, T157602, was reported as an HSV-2 HPI [6] and thereafter two HPIs, BILS 22 BS and Bay 57-1293 (pritelivir), were shown to have anti-HSV activity in vitro and therapeutic activity in murine HSV models [7,8,9]. Next, three classes of herpesvirus HPIs were developed: thiazole urea (pritelivir [9]), 2-amino-thiazolylphenyl derivatives (BILS 179 BS [8]), and oxadiazolylphenyl type (ASP2151, amenamevir) [10]. Amenamevir has anti-HSV and anti-VZV activity, while, in contrast, pritelivir and BILS 22 BS have anti-HSV activity but lack anti-VZV activity. Clinical studies of HPIs on genital herpes using amenamevir and pritelivir have been conducted [11,12,13]. 

We characterized the profile of the anti-HSV and anti-VZV activities of amenamevir [14,15,16,17,18]. The antiviral activity of amenamevir was not influenced by viral DNA synthesis, although that of acyclovir was attenuated [17,18]. Amenamevir was licensed for the treatment of herpes zoster on the basis of a clinical trial on herpes zoster treatment in September 2017 [19]. In this review, we focus on the anti-HSV and anti-VZV activities of amenamevir and discuss the differences in its antiviral activity compared with acyclovir. 

## 2. Role of HP in DNA Synthesis

Double-stranded DNA needs to be separated into two single strands (replication fork) for DNA synthesis, and these two separated complementary strands proceed to form two new double-stranded DNA molecules during DNA replication (Figure 1). HP is an essential enzyme complex for DNA synthesis conserved from *Escherichia coli* to *Homo sapiens*. HSV, VZV, and host cells produce specific HP as an essential and indispensable gene product related to their DNA replication.

The HP complex possesses multiple enzymatic activity, including DNA-dependent ATPase, helicase, and primase activities, all of which are required for the functions of HP in viral DNA replication. The helicase relaxes the duplex DNA ahead of the replication fork and separates the double strand into two single strands. The primase sets RNA primers for DNA synthesis on the single-stranded DNA and the DNA polymerase extends the DNA (Okazaki fragment) as a new complementary DNA following the RNA primer in the lagging strand. Okazaki fragments are ligated and DNA synthesis in the lagging strand is completed. 

The HP enzyme complex of herpesviruses consists of three proteins—UL5 (helicase, VZVORF55), UL52 (primase, VZVORF6), and UL8 (cofactor, VZVORF52)—which are well conserved among the Herpesviridae. Helicase unwinds duplex DNA ahead of the fork and separates the double strand into two single strands. Primase lays down RNA primers that the DNA polymerase complex (UL30/UL42) extends. The HP complex possesses multi-enzymatic activities mediated by DNA-dependent ATPase, helicase, and primase. 

## 3. HP Inhibitors of HSV

HPIs bind to the helicase-primase complex and inhibit single-stranded, DNA-dependent ATPase, helicase, and primase activities [10,14,20,21]. T157602 was first reported as an HPI of the HSV UL5-UL8-UL52 complex with the use of a high-throughput biochemical DNA helicase assay [6]. T157602 inhibited helicase activity, primase activity, and the replication of HSV types 1 and 2 without cytotoxicity. Seven independently isolated T157602-resistant mutant viruses (four HSV type 2 and three HSV type 1) carried a single base pair mutation in UL5 that resulted in a single amino acid change in the UL5 protein. When the mutated UL5 gene from T157602-resistant HSV was transferred to a sensitive HSV, it acquired T157602 resistance. 

Three HPIs, pritelivir, BILS 179 BS, and amenamevir, have anti-HSV activity, whereas amenamevir alone has anti-VZV activity [8,9,10,22]. Amenamevir was more effective than valacyclovir for treating HSV skin lesions in HSV mutant-infected mice, and HPI-resistant HSV mutants were susceptible to acyclovir with attenuated growth in vitro and reduced pathogenicity compared with the parent virus [14]. Mutations in the helicase or primase of the HP complex against amenamevir impaired viral replication and pathogenicity. Amenamevir showed better efficacy than valacyclovir at treating HSV zosteriform skin lesions in immunocompromised mice [23]. 

## 4. Comparison of Anti-Herpes Virus Activity in Three HPIs

Figure 2 shows the structures of four HPIs. These HPIs are virus-specific with low cytotoxicity in vitro. Pritelivir and amenamevir are orally available, effective against HSV infection, and are well tolerated in mice and humans. The target molecules of HPIs are different from acyclovir, penciclovir, foscarnet, and vidarabine; therefore, their mechanism of action and antiviral and pharmacokinetic profiles are unique to HPIs. HPIs have a lower EC_50_ for HSV compared with acyclovir. Furthermore, the anti-VZV activity of amenamevir is markedly different from the other anti-herpetic HPIs. The EC_50_ of HPIs for HSV-1 and HSV-2 were low (0.014–0.060 µM and 0.023–0.046 µM, respectively) and amenamevir (0.038–0.10 µM) was more potent against all VZV strains tested compared with acyclovir (1.3–5.9 µM) [10]. Only amenamevir has anti-VZV activity with a similar IC_50_ value to HSV; however, pritelivir and BILS 22 BS had 100–200 times higher IC_50_ values than amenamevir. Amenamevir has a unique antiviral spectrum for HSV and VZV. Furthermore, it had efficacy against HSV-1, HSV-2, and acyclovir-resistant/TK-deficient virus infection and exhibited synergistic activity against HSV-1 and HSV-2 with acyclovir and valacyclovir in vitro and in vivo, respectively [3,16,24]. 

Amenamevir had a better antiviral activity profile during the DNA synthesis phase compared with acyclovir when used as an anti-HSV and anti-VZV drug, and compared with acyclovir, the anti-VZV and anti-HSV activities of amenamevir were not affected by viral DNA synthesis in the infected cells [17,18]. Acyclovir is effective against infected cells after infection and before viral DNA synthesis, but it is not effective against infected cells with abundant viral DNA synthesis. Conversely, amenamevir is equally effective against infected cells immediately after infection or in the late phase of infection, when viral DNA synthesis is abundant. This suggests that amenamevir would have high efficacy in treating severe VZV and HSV infections with a high viral load.

## 5. Amenamevir Resistance

Amenamevir-resistant viruses have been isolated, and sequencing analyses revealed several single-base-pair substitutions resulting in amino acid changes in the helicase and primase of amenamevir-resistant HSV mutants [10]. Amino acid alterations in the helicase subunit were clustered near helicase motif IV in the UL5 helicase gene of HSV-1 and HSV-2, whereas the primase subunit substitution was found only in amenamevir-resistant HSV-1 mutants. The combined mutation of R367H and S364G in the UL52 primase gene affords greater resistance to amenamevir than the S364G mutation alone. The accumulation of mutations increases the resistance to amenamevir, as shown in Table 1. Amenamevir-resistant HSV mutants had reduced growth capability in vitro and pathogenicity compared with the parent virus in HSV-infected mice [14]. 

Because the HP is essential for virus growth, an HP-deficient virus cannot replicate, in contrast to a TK-deficient virus. Any mutation in the thymidine kinase gene resulting in TK deficiency or reduced activity becomes an acyclovir-resistant mutant. Amenamevir-resistant viruses can replicate in the presence of amenamevir by avoiding interactions with HP. Acyclovir-resistant mutants of HSV and VZV with TK deficiency are similarly as susceptible as wild-type strains to amenamevir (Table 2) [16]. Thus, acyclovir-resistant HSV and VZV are susceptible to amenamevir, and this is consistent with the fact that the targets of acyclovir and amenamevir are thymidine kinase and helicase-primase, respectively, with no cross resistance. Accordingly, amenamevir-resistant HSV was as susceptible to acyclovir and penciclovir as the wild-type strain. 

The mean frequencies of amenamevir- and acyclovir-resistant HSV variants were 1.19 × 10^−6^ and 1.65 × 10^−3^, respectively, from four or two strains of wild HSV-1 and HSV-1 virus stocks, respectively, and the frequency of pre-existing amenamevir-resistant HSV variants was 1389 times lower than that of acyclovir-resistant variants [14]. The frequency of acyclovir-resistant mutants in the virus stock was approximately 1 in 1000 plaque forming units. Amenamevir-mutants should preserve the function of HP to replicate carrying the restricted amino acid change in UL5 or UL52 and the frequency of amenamevir-resistant virus was approximately 1 in 10^6^ plaque-forming units [10,14]. Thus, amenamevir resistance is lower than acyclovir resistance, which suggests that the effectiveness of amenamevir is partly related to the lack of amenamevir-resistant strains. 

Amenamevir-resistant mutants had attenuated growth and pathogenicity compared with the wild-type strain [14]. Furthermore, amenamevir had antiviral activity against VZV [10,14,15,16].

## 6. Nucleotides and DNA Synthesis

Figure 3 illustrates nucleotide metabolism and the viral enzymes involved in viral DNA synthesis. Purines and deoxyribonucleotide triphosphates (dXTP) are synthesized from amino acids to inosine-monophosphate (IMP) and then to the ribosyl form of adenosine-MP (rAMP), guanosine-MP, cytosine-MP, and uridine-MP. Ribonucleotide-MP are in the form of ribose (rNMP: RNA type), and their triphosphate forms (rNTPs) are substrates for RNA. The ribose forms of ribonucleotide diphosphate (rNDP) are then converted to the deoxyribose forms of dNDPs by ribonucleotide reductase (RR) and further to dNTP to become the substrates for DNA. Uridine diphosphate (dUDP) is converted to the monophosphate forms dUMP and dUMP, substrates for thymidylate synthase (TS), which are converted into thymidine-MP (dTMP) and then successively to dTTP for DNA synthesis. 

Basal cells in the skin synthesize DNA for cell proliferation and differentiation to keratinocytes; however, most skin cells do not synthesize DNA, and instead they synthesize RNA and proteins to maintain cellular functions. Because skin cells contain low amounts of deoxyribonucleotides for DNA synthesis, when HSV or VZV infection induces thymidine kinase, acyclovir or penciclovir is phosphorylated and further converted to acyclovir or penciclovir triphosphate, which results in the efficient inhibition of viral DNA synthesis by chain termination. When viral RR and TS are induced, rNTP, a substrate for RNA, is converted to dNTP, which increases dGTP. dGTP is increased in HSV-infected cells at 4 h after infection. Therefore, HSV and VZV traffic between infected cells to facilitate viral DNA synthesis by converting RNA synthesis to viral DNA synthesis via the induction of enzymes in the newly infected cells [18]. One molecule of the VZV and HSV genome contains 60,000 and 90,000 guanosines, respectively, and abundant dGTP competes with acyclovir- or penciclovir-TP for DNA polymerase, which results in the inefficient inhibition of viral DNA synthesis, when viral DNA synthesis is abundant, as shown in Figure 4.

## 7. Antiviral Activity of Amenamevir Is Not Influenced by the Replication Cycle

Cells infected with viruses are treated with antiviral drugs immediately after infection, and the drug concentration at 50% plaque formation of the inoculated virus is expressed as the EC_50_. After infection, cells will be at various stages of viral DNA synthesis (immediately after infection and prior to viral DNA synthesis, viral DNA synthesis initiated, and production of viruses during viral DNA synthesis). Therefore, it might be possible to determine the actual concentrations of antiviral drugs required to inhibit virus growth in vivo by examining the susceptibility of infected cells to antiviral drugs at various stages after virus infection. We investigated the susceptibility of infected cells to acyclovir and amenamevir after infection with HSV and VZV.

The time course of the antiviral activity of amenamevir and acyclovir after HSV and VZV showed contrasting profiles related to the concentration required for the 50% plaque reduction of infected cells (Figure 4). The EC_50_ values of acyclovir were increased 5.0–7.5 h after HSV infection and reached approximately 10 and 8 times the EC_50_ of 0 h at 12.5 h in contrast to ASP2151 and foscarnet [17,18]. The EC_50_ values of amenamevir and foscarnet were not affected 0 to 12.5 h after HSV infection. The EC_50_ values of acyclovir and sorivudine increased 6 h after VZV infection and reached approximately 10 times the EC_50_ of 0 h at 18 h [18]. 

Conversely the EC_50_ values of amenamevir and foscarnet were not affected 0 to 18 h after VZV infection. The increase in the EC_50_ values of infected cells to acyclovir began at the time of HSV and VZV DNA synthesis and the effects of viral DNA synthesis and its related cellular events clearly influenced the antiviral activity of acyclovir. The antiviral activity of amenamevir was not affected by the replication cycle of VZV and HSV, whereas the late phase of infected cells was 10 times less susceptible to acyclovir than immediately after infection [17,18]. 

The susceptibility of infected cells to acyclovir after viral DNA synthesis was decreased because of the increased amount of dGTP present for viral DNA synthesis in the late phase of viral replication (Figure 4 and Figure 5). Acyclovir triphosphate competes with dGTP for viral DNA polymerase in infected cells, and when the dGTP supply becomes abundant, anti-HSV activity is attenuated by competing with acyclovir triphosphate, which results in an increase in the EC_50_ values of acyclovir. Viral RR converts the ribose form of guanosine diphosphate (rGTP) to the deoxyribose form of deoxyguanosine diphosphate (dGDP), which is converted to dGTP, a substrate for viral DNA polymerase. Hydroxyurea is an inhibitor of RR that inhibits the supply of dGTP. Acyclovir and hydroxyurea treatment did not reduce the acyclovir susceptibility of HSV-infected cells 12.5 h after viral DNA synthesis or of cells without hydroxyurea dGTP supplied by viral RR, which competes with acyclovir triphosphate, which decreases the anti-HSV and anti-VZV activity of acyclovir in the late phase of cells infected with HSV and VZV [17,18,25]. Conversely, HPIs target the HP and not DNA polymerase, and the nucleoside analog is not a substrate of the HP. The anti-HSV activity of amenamevir was not influenced by the time course of infection, the status of infected cells, or the replication cycle of the virus, which is a major advantage of the HPIs over the current anti-herpetic drugs, acyclovir, valacyclovir, and famciclovir.

## 8. Synergism of Amenamevir with Other Antiherpetic Drugs

We analyzed the antiviral interactions of amenamevir with acyclovir and penciclovir when used to treat HSV and VZV infection by isobologram in a plaque reduction assay using the response surface model. The combination of amenamevir with acyclovir had statistically significant synergistic antiviral activity against the tested strains of HSV-1, HSV-2, and VZV (*p* < 0.0001, *p* = 0.0009, *p* = 0.0005, respectively) [15]. The antiviral activity of amenamevir combined with acyclovir and penciclovir against wild-type HSV-1, HSV-2, and VZV demonstrated statistically significant synergistic activity at all concentrations (*p* < 0.05) (Figure 6). Amenamevir with vidarabine showed additive effects against wild-type HSV-2 and synergistic effects against VZV. Low concentrations of amenamevir had stronger synergism with acyclovir or penciclovir compared with higher concentrations of amenamevir in the isobologram analysis for HSV-1, HSV-2, and VZV. 

The increased efficacy of amenamevir at lower concentrations with acyclovir and penciclovir for HSV and VZV indicated that amenamevir might reduce the number of viral DNA replication forks and allow acyclovir to inhibit the reduced sites of viral DNA synthesis. This is an important pharmacological issue for combination therapy with amenamevir and acyclovir. A single dose of amenamevir maintains an antiviral blood level throughout the day, which reduces the number of replication forks and DNA synthesis sites. Thus, an effective concentration of amenamevir is effective in cells immediately after infection and cells in which viral DNA synthesis is abundant. However, acyclovir is effective for cells in the early stage of infection, but not for cells with abundant viral DNA synthesis. Amenamevir and acyclovir are effective for cells in the early stages of infection. Although acyclovir is not effective in cells with abundant viral DNA synthesis, amenamevir inhibits viral DNA synthesis by reducing the number of forks and reducing the number of replication forks susceptible to acyclovir, even in cells with abundant viral DNA synthesis. Amenamevir and acyclovir not only have synergism in cells in the early stage of infection, as shown by the plaque reduction method, but also have efficacy on cells with advanced infection and abundant viral DNA synthesis.

Synergism of amenamevir and valacyclovir related to antiviral activity was examined in a mouse HSV zosteriform model, and the inhibition of progression of zosteriform lesions by combination therapy was more potent than that of either drug as a monotherapy [15]. The efficacy of amenamevir was not affected by the host’s immune status in terms of effective oral doses in immunocompromised mice [23]. Amenamevir was effective at treating severe skin infections, even when the start of treatment was delayed, whereas valacyclovir was ineffective.

These results indicate that combination therapies of amenamevir with acyclovir have synergistic anti-herpes effects against HSV and VZV infections in vitro and in vivo. Therefore, the combination of amenamevir with acyclovir may be a useful approach to treat herpes infections and might be a more effective therapeutic option than monotherapy for the treatment of herpes encephalitis or immunosuppressed patients.

In severe VZV and HSV infections, where viral replication is abundant, the susceptibility to acyclovir is low because of DNA synthesis and the effect is unlikely, as shown in Figure 5. However, amenamevir is not affected by viral DNA synthesis, even if viral DNA synthesis is abundant. Moreover, because replication fork formation is restricted by the action of amenamevir, acyclovir at low concentrations inhibits DNA synthesis in the presence of amenamevir, as shown in Figure 6. Therefore, combination therapy that takes advantage of the characteristics of acyclovir and amenamevir is recommended for severe VZV and HSV infections.

## 9. Pharmacokinetic Advantage of Amenamevir

Acyclovir and penciclovir are excreted in the urine as renal excretory drugs, and their oral administration two or three times a day is necessary to maintain a drug concentration in the blood that preserves their antiviral activity over a whole day. Administration of 1000 mg of valacyclovir reached 5.65 ± 2.37 μg/mL of acyclovir in the serum, with an elimination half-life of 3.03 ± 0.13 h. The concentration decreased to 2 μg/mL or less within 4 h (Weller et al., 1993). The EC_50_ of VZV-infected cells was 0.745 μg/mL at 0 h after infection, >2 μg/mL at 6 h, and after 6 h, acyclovir did not exhibit sufficient anti-VZV activity against VZV-infected cells (Figure 4) [18]. 

Approximately 75% of amenamevir is excreted in the feces and 20% in the urine. Amenamevir is not a renal excretion type drug and therefore its blood concentration can be maintained for a long time. A single dose of 300 mg of amenamevir preserved a mean plasma concentration over 9 times higher than the EC_50_ after 24 h [26]. Plasma amenamevir concentrations required to completely suppress HSV-1 growth were seven times higher than the EC_50_ in a mouse skin infection model [27]. A once-daily dose can maintain an antiviral concentration for 24 h, and this pharmacokinetic profile is longer than that of renal excretion type drugs, such as acyclovir and penciclovir, especially for recurrent genital herpes. Suppressive therapy with valacyclovir, famciclovir, or acyclovir successfully suppresses recurrent episodes but they do not maintain an effective antiviral concentration, which allows asymptomatic viral shedding and transmission. HPIs can maintain an antiviral concentration for a whole day, thereby completely suppressing viral replication, including the complete inhibition and viral shedding of genital herpes. HPIs are expected to demonstrate their true value as an antiviral for the suppressive treatment of recurrent genital herpes.

## 10. The Fate of HSV- and VZV-Infected Cells

Immune responses to VZV and HSV consist of innate immunity and adaptive immunity. Furthermore, cell-mediated immunity specific to VZV and HSV causes eruptions and vesicles in the skin. Erythema multiforme occurred at the site of apparently normal skin recovered from HSV skin lesions 1–3 weeks after infection and HSV DNA was detected in the cells of erythema multiforme lesions [28]. Although VZV is strongly cytolytic in cell cultures, the process by which infected cells become apparently normal cells with viral DNA was reproduced in cell culture by antigenic modulation with an anti-gH neutralizing antibody [29]. The skin lesions caused by HSV and VZV reverted to apparently normal skin over time without causing erosions/ulcers or skin defects related to extensive cell necrosis. Infected cells that have completed viral DNA synthesis are resistant to acyclovir (Figure 4) and continue to survive and express viral antigens. These cells become the target of the immune response until returning to normal.

## 11. Innate and Adaptive Cell-Mediated Immunity Related to the Clinical Image of Viral Infection

Typical inflammation caused by adaptive immune responses characterized by cell-mediated immunity is termed delayed type hypersensitivity (DTH) or type IV hypersensitivity—tuberculin tests, contact dermatitis, or urushiol-induced dermatitis are examples. DTH occurs as redness and swelling at 5–6 h and peaks at 48–72 h after contact with an antigen, and even in the absence of the initial antigen, which then resolve 1 week later.

Photodistribution revealed the role of innate immune responses to HSV and VZV (Figure 7) [30]. Sunburn (ultraviolet light) allows virus growth before inducing a phase of adaptive immunity by impairing innate immunity related to the function of Langerhans cells in the skin. This results in more severe skin lesions in the sunburned skin compared with non-sunburned skin in the adaptive immunity phase [31]. 

Skin lesions are generally only present in areas where viral spread is not suppressed by innate immunity. However, ultraviolet rays impair innate immune responses, especially those mediated by Langerhans cells, which allows the initial viral infection to spread, which results in broader, denser, and more severe skin lesions generated by an adaptive cell-mediated immune response termed photodistribution. The extent of viral lesions is determined by the size of the virus replicated area before adaptive cell-mediated immune responses are induced. This demonstrates that innate immunity has an important role in the distribution and density of viral skin lesions by suppressing viral spread. 

This is consistent with the idea that early treatment before the appearance of lesions alleviates skin lesions in varicella and recurrent genital herpes. Treatment with acyclovir in the latter half of the incubation period of chickenpox resulted in subclinical or mild varicella [32]. Furthermore, for recurrent genital herpes, treatment at the prodromal stage before the appearance of skin lesions blocks their appearance in one-third of patients [33,34]. If antiviral agents are administered before the adaptive cell-mediated immune response is initiated, they can prevent virus localization, distribution, and replication, which results in mild or subclinical skin lesions and alleviated disease. This concept of prophylactic or preemptive therapy was achieved in cytomegalovirus (CMV) pneumonia in immunocompromised patients by starting ganciclovir treatment before apparent CMV replication appeared [35,36].

Anti-inflammatory steroids reduced inflammation of urushiol-induced dermatitis caused by adaptive cell-mediated immunity (DTH), but did not shorten the duration of inflammation. Similarly, antiviral drugs prevented the appearance of new skin lesions and halted the progression to vesiculation, but did not shorten the time to the elimination of inflammation, even when prednisolone was administered [19,37,38]. Inflammation appeared 3 days after contact with a viral antigen and inflammation induced by herpes zoster was exacerbated 3 to 5 days after the onset of inflammation by the continuous presence of infected cells and continued for 3 weeks (Figure 7). The infected cells continued to receive antigen stimulation and inflammation peaks 3 to 5 days after the appearance of the rash. Antiviral drugs halted the appearance of new lesions and the enlargement of skin lesions but did not suppress inflammation once it was induced. The inflammatory process continued for 3 weeks before being resolved, even under the intervention of amenamevir treatment initiated on day 2 of herpes zoster infection (Figure 7).

## 12. Timing of Antiviral Therapy

The effectiveness of antiviral drugs can be maximized by limiting the spread, distribution, and size of viral infections before the onset of adaptive cell-mediated immunity as observed in photodistribution. Therefore, antiviral therapy should be started in the prodromal period when innate immunity is available or as soon as VZV and HSV infections are diagnosed (Figure 8).

Anti-VZV drug treatment for varicella is started within 24 h after the onset of symptoms and continues for 7 days. However, eruptions do not always progress to vesiculation, ending in an abortive form of infection. Varicella is highly contagious and exposure to it causes approximately 80% of infections within a family. After exposure to varicella, family members were prophylactically treated with acyclovir during the first and second halves of the incubation period of about 14 days, and infection was assessed by an increase in antibody titers [32]. In the first half group of 11 patients, varicella developed in 91% of cases and 9% was subclinical, whereas in the second half group of 11 patients, a very mild disease occurred in 27% of cases and subclinical disease developed in 73%. Prophylactic administration converted overt varicella to an asymptomatic infection in the second half of the incubation period. Acyclovir inhibited viral replication in the skin when administered immediately before the onset of varicella (corresponding to the prodromal period), which resulted in subclinical or mild disease. Although prophylactic acyclovir treatment was convenient and useful, the introduction of universal varicella vaccination has superseded it [39,40].

Herpes zoster is uncommon with an annual incidence of approximately 1 in 100 people aged > 60 years old [41,42]. Seventy to eighty percent of patients with herpes zoster experience prodromal symptoms, such as burning, shooting, stabbing, or throbbing pain in the dermatome(s), which represent allodynia [43,44]. Sclerotomal pain usually precedes dermatomal pain by a few days in the prodrome [45]. Although the preemptive use of antiherpetic drugs at the prodromal symptom stage might be the optimal timepoint to alleviate the severity of herpes zoster, it is not easy to differentially diagnose the prodrome of herpes zoster from other causative symptoms.

Recurrent genital herpes causes the appearance of unpleasant vesicles, erosions, and ulcers that last for approximately 1 week. Prodromal treatment with anti-HSV drugs prevents uncomfortable genital lesions in up to one third of patients and helps to maintain a comfortable daily life [33,34]. The oral administration of acyclovir, valacyclovir, and famciclovir starting within 24 h of the onset of recurrent herpes and continuing for a period of 5 days is effective at reducing the duration of symptoms by a median of 1–2 days. 

CMV pneumonia was treated with ganciclovir in transplant recipients and immunocompromised patients in the 1980s, and current guidelines have been established to prevent the development of refractory CMV pneumonia by the administration of prophylactic or preemptive treatment with ganciclovir or letermovir [35,36,46]. The optimal timing for the treatment of herpes virus diseases is prodromal and preemptive therapies with antiherpetic drugs block the onset of diseases such as varicella, herpes zoster, genital herpes, and CMV pneumonia. Although it is difficult to diagnose the prodromal stage in VZV infections, it is easier for HSV and CMV infections. The optimal timepoint for the antiviral treatment of VZV infection is at least within 24 h for varicella and within 72 h for herpes zoster, as soon as possible after the onset or diagnosis.

## 13. Clinical Trials of Herpes Zoster Treatment with Amenamevir

Current suppressive therapies for genital herpes using acyclovir, valacyclovir, and famciclovir are effective, but these drugs do not provide efficacious antiviral concentrations over the whole day because they are renal excretory drugs [26], and this allows viral shedding and viral replication.

On the basis of the promising preclinical profiles related to the antiviral activity and pharmacokinetics of HPIs, pritelivir and amenamevir were evaluated in two phase 2 clinical studies of patients with genital herpes [11,12,13]. HPIs have an excellent pharmacokinetic profile, and when administered daily, they maintain anti-HSV activity for the whole day, which results in the inhibition of HSV replication, reactivation, and shedding, and lesion formation in patients with genital herpes, as well as the transmission of HSV from asymptomatic and symptomatic cases of genital herpes. Although one of the indications for HPI treatment is the suppressive treatment of genital herpes, sufficient clinical efficacy has not been achieved, and they have not replaced current suppressive therapies. 

The efficacy and safety of amenamevir 400 mg once daily were evaluated in a phase 3 randomized, double-blind, valacyclovir-controlled phase 3 study when compared with valacyclovir 1000 mg three times daily in 751 Japanese patients with herpes zoster within 72 h after the onset of rash [19]. The proportion of cases with reduced new lesion formation by day 4 (primary efficacy endpoint) was 81.1% (197/243) for amenamevir and 75.1% (184/245) for valacyclovir and the non-inferiority of amenamevir to valacyclovir was confirmed using a closed testing procedure. Furthermore, 10.0% (25/249) and 12.0% (30/249) of patients receiving amenamevir or valacyclovir, respectively, experienced drug-related adverse events. Days to cessation of new lesion formation, complete crusting, healing, pain resolution, and virus disappearance (secondary endpoints) were not statistically different between the amenamevir and valacyclovir groups.

Although amenamevir has excellent pharmacokinetics and anti-VZV properties compared with acyclovir, it did not show clinically superior efficacy over valacyclovir in immunocompetent subjects with herpes zoster [19]. Sorivudine 40 mg once daily had better efficacy against herpes zoster than acyclovir 800 mg five times daily in patients infected with human immunodeficiency virus [47,48]. The target period of the administration of anti-VZV agents to inhibit viral replication in the skin of immunocompetent subjects with herpes zoster might be limited to a few days, but longer in immunocompromised patients, which might demonstrate the beneficial action of amenamevir over acyclovir in immunocompromised subjects. Amenamevir (Amenalief^®^) has been approved as an anti-herpes zoster drug and has been successfully used to treat approximately 1,240,000 herpes zoster patients in Japan.

## 14. Post Marketing Surveillance

At the time of approval in Japan, “renal disorder,” “cardiovascular event,” and “platelets decreased” were set as important potential risks in the Japanese risk management plan (RMP) to be monitored during routine pharmacovigilance (PV) activities. As of June 2020, “erythema multiform” was added as an important identified risk, and “toxic epidermal necrolysis” and “Stevens–Johnson syndrome” were added as important potential risks into the Japanese RMP.

As an additional PV activity to further evaluate and demonstrate the efficacy and safety of amenamevir in herpes zoster patients in a clinical setting, data were collected by a Special Drug Use Surveillance program (an observational study, protocol No. AME11) [49]. This study specifically collects information on the efficacy and safety of amenamevir in routine clinical practice in patients who use the drug for herpes zoster. In addition, pain status was followed to examine post-herpetic neuralgia. This study plans to enroll 3000 patients; as of 2 January 2019, 1446 patients have been enrolled. The safety analysis set includes 1346 patients, of which 11 experienced adverse drug reactions, with an incidence of 0.82% (11/1,346 patients). Among the 11 reported cases of adverse drug reactions, two patients had abdominal pain and two had diarrhea and fever. Regarding safety issues related to the important potential risks identified in the Japanese RMP, one case of thrombocytopenia and one case of gingival bleeding were reported as events associated with “platelets decreased”, and one case of palpitation was reported as an event associated with “cardiovascular event”. None of these were serious.

In Japan, everyone has health insurance, which covers most patients with shingles who are treated at medical institutions. The estimated number of patients with herpes zoster treated with Amenalief^®^ from its launch in September 2017 to February 2021 is approximately 1,240,000 based on sales data and the estimated dose per patient in Japan. From estimations using the number of prescriptions in 2020, the total number of new herpes zoster patients in Japan is approximately 940,000 per year, of which 240,000 (25.5%) are treated with amenamevir.

## 15. Cumulative Number of Adverse Drug Reactions in Post-Marketing

Table 3 shows the number of adverse drug reactions related to oral anti-herpetic drugs (valaciclovir, acyclovir, famciclovir, and amenamevir) in the database (Japanese Adverse Drug Event Report database) maintained by the Pharmaceuticals and Medical Devices Agency (PMDA), the Japanese regulatory authority. Overall, 500–700 adverse drug reactions related to oral anti-herpetic drugs are reported every year, and most events are categorized as “nervous system disorders” and “renal and urinary disorders.” The incidence of adverse drug reactions is not calculated, because they are spontaneous reports from healthcare providers or consumers to a pharmaceutical company and/or regulatory authority such as the PMDA. Because the Ministry of Health, Labor and Welfare of Japan requires the widespread reporting of cases with suspected side effects/adverse events, each report states that there is a causal relationship between the drug and the symptoms and abnormal findings. Furthermore, the same case may be reported by multiple reporters. All reported cases are listed, but the relationship with the drug has not been evaluated and should be interpreted with caution. Accordingly, it is not possible to simply evaluate or compare the safety of drugs based on the number of cases in the case reports or the number of cases in the list of reported adverse drug reactions. However, the information obtained here includes data that cannot be obtained by the post-marketing surveillance of amenamevir, although our interpretation should be carefully evaluated with the understanding of the limitations of the above information. These events are a medical problem for current nucleoside analogs such as valaciclovir, acyclovir, and famciclovir in clinical practice. The type and the number of events related to amenamevir are shown in parenthesis (Table 3).

## 16. Amenamevir against Herpes Zoster Caused by Acyclovir-Resistant VZV

A 64-year-old male patient with adult T-cell lymphoma with stem cell transplantation suffered herpes zoster and was treated with acyclovir. VZV was isolated from 12 vesicles and the susceptibility of the 12 isolated viruses from each vesicle was examined to acyclovir and amenamevir 14 days after acyclovir treatment was investigated. Half of the 12 isolated viruses were a mixture of acyclovir-susceptible and acyclovir-resistant viruses with reduced susceptibility to acyclovir compared with the wild-type strain, which indicates that the isolated viruses were in the transition phase of susceptible VZV to resistant VZV harboring mutations (p.Phe139Serfs*25, p.Try144Phefs*20) in the thymidine kinase gene. Amenamevir treatment quickly terminated the appearance of new vesicles that subsequently scabbed over 10 days later [50]. Furthermore, amenamevir showed efficacy against acyclovir-resistant VZV infection in one case, as expected from the finding that amenamevir has efficacy against acyclovir resistance in vitro (Table 2) [15,16].

## 17. Conclusions and Overall Perspectives

HPIs have been developed as new anti-herpes drugs, but currently only amenamevir among the HPIs is used for the treatment of herpes zoster. Amenamevir has a low EC_50_ to HSV and VZV and its efficacy in HSV-infected animals and synergism with acyclovir and penciclovir was indicated for the combinational treatment of severe infection with HSV or VZV [14,15,16,17,18]. The target enzymes of amenamevir and acyclovir are different—amenamevir is effective against acyclovir-resistant viruses and has been used successfully to treat acyclovir-resistant VZV in a patient with herpes zoster caused by acyclovir-resistant VZV [50]. Amenamevir (Amenalief^®^) has been approved as an anti-herpes zoster drug and has been successfully used to treat approximately 1,240,000 herpes zoster patients in Japan. To date, the numbers of adverse reactions in amenamevir-treated patients seem to be lower than for other anti-herpetic drugs (Table 3). 

The relationship between viral replication in HSV and VZV infections and innate and adaptive immunity is explained by the pathophysiology of photodistribution, which affects the distribution of skin lesions, as stated in Section 11. The optimal treatment timepoint is at the prodrome during photodistribution, which prevents the formation of skin lesions. This is supported by clinical observations that antiviral treatment in the latter half of the latent period of chickenpox and prodromal treatment in recurrent herpes can prevent the appearance of skin lesions [32,33,34]. However, it is difficult to start treatment for chickenpox and herpes zoster from the prodrome; therefore, it is best to start treatment immediately after the onset and diagnosis. Vesiculation of herpetic lesions can be prevented by the early treatment of HSV skin lesions and chickenpox as an abortive form. Treatment after the onset of herpes zoster can stop the appearance of new skin lesions and prevent its spread after 4 days, but it has no effect on the exacerbation of inflammation induced by adaptive immunity, and the inflammation is augmented even after the start of antiviral treatment. Approximately 3 weeks of inflammation cannot be shortened by the use of anti-inflammatory drugs. As described above, the efficacy of antiviral drugs and inflammation caused by adaptive immune responses should be evaluated separately because viral replication and generating adaptive immunity are independent pathophysiologic events. Therefore, antiviral agents can stop viral spread but cannot alleviate adaptive immunity, which exacerbates inflammation after the start of antiviral therapy. Before starting antiviral treatment, it is recommended to explain the following to the patient: inflammation related to lesions already present will worsen for the next 3 days, which is unavoidable because it is an immune response, but antiviral drugs will alleviate herpes zoster by stopping viral replication and limiting the extent of the lesion and preventing the spread of the infection. 

The long-lasting antiviral concentration of HPIs can inhibit HSV replication throughout the whole day, which prevents the reactivation of HSV from ganglia, subsequent viral shedding, and its sexual transmission. The excellent pharmacokinetic profile of HPIs for the suppressive therapy of genital herpes indicates that they might provide greater benefit than current therapy with valacyclovir and famciclovir, although this has not been reported to date. Thus, HPIs might have favorable characteristics as antiherpetic drugs for suppressive therapy.

The clinical efficacy and safety profiles of amenamevir have been established in patients with herpes zoster, which indicates that amenamevir as an HPI might be an anti-herpes therapy. This review introduced the newly developed HPI, amenamevir, and suggests that HPIs might be next-generation drugs for HSV and VZV infections.

## Figures and Tables

**Figure 1 viruses-13-01547-f001:**
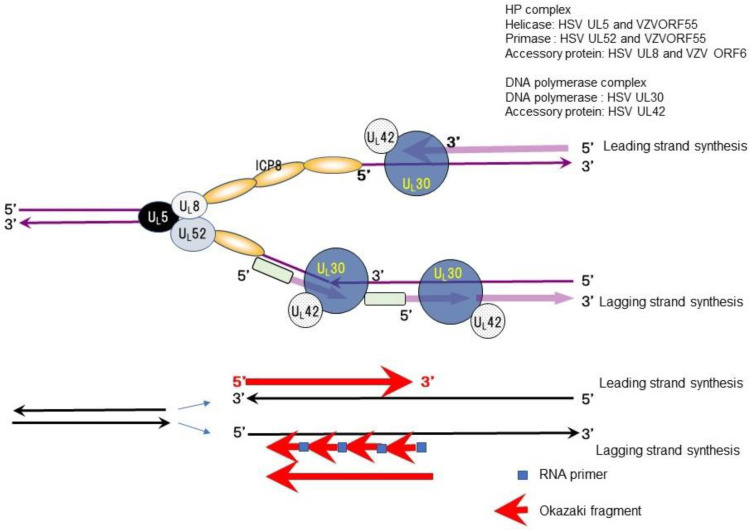
The viral helicase primase (HP) complex in viral DNA synthesis (modified from [5]). The figure shows the role of the HP complex (UL5, UL8, and UL52 of HSV and ORF55, ORF6, and ORF52 of VZV), the DNA polymerase complex (UL42 and UL30 of HSV and ORF28 of VZV: DNA polymerase), and the ICP8 single-stranded DNA binding protein of HSV (ORF29 of VZV). HSV UL5 and VZVORF55 (helicase) relax double-stranded DNA and separate double strands into two single strands, forming a replication fork. HSV UL52 and VZVORF55 (primase) synthesize RNA primers (followed by Okazaki fragments) for lagging strand DNA synthesis. DNA polymerase contains intrinsic ribonuclease H (RNase H) activity that specifically degrades RNA/DNA heteroduplexes formed from RNA primer-Okazaki fragments and the template DNA complex. DNA polymerase and its accessory protein (UL42) bind to each single strand and synthesize complementary DNA to each strand of the replication fork. The single-stranded DNA binding protein with helix destabilizing activity, ICP8 (UL29 of HSV and VZV), binds to a single-stranded template DNA with helix destabilizing activity. The arrows indicate the direction of movement of the DNA replication proteins.

**Figure 2 viruses-13-01547-f002:**
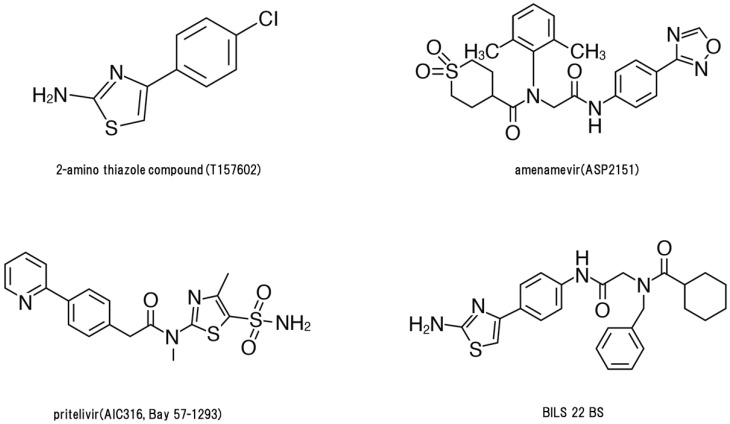
Structures of helicase-primase inhibitors.

**Figure 3 viruses-13-01547-f003:**
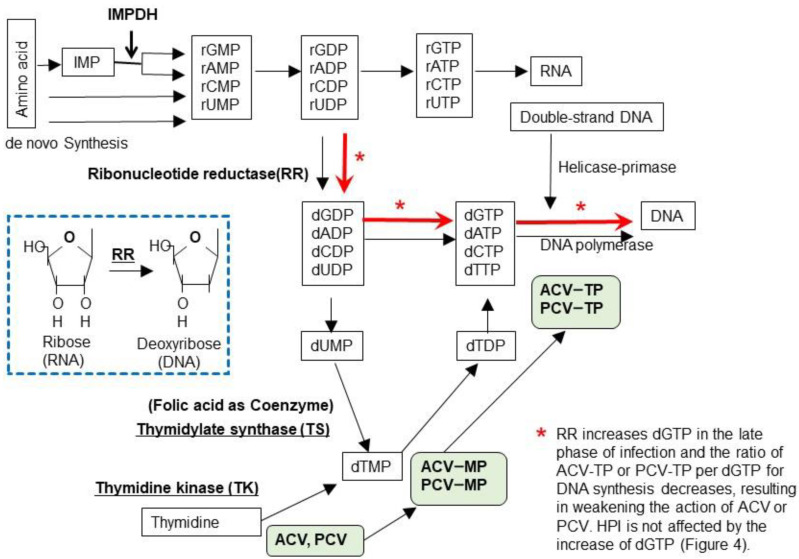
Biosynthesis of nucleotides. Purines and pyrimidines are synthesized de novo from amino acids as the ribose form of nucleotides and inosine monophosphate (IMP) that are modified by IMP dehydrogenase to adenosine-monophosphate (rAMP) and guanosine-monophosphate (rGMP). Next, nucleotide monophosphate (rNMP) is phosphorylated to a triphosphate form (rNTP), which is the substrate for RNA. The ribose form of nucleotide diphosphate (rNDP) is converted to the 2’-deoxyribose form (dNDP) by cellular or viral ribonucleotide reductase (RR), as shown in the lower box. When viral RR is induced by HSV and VZV infection, dNDPs are synthesized in the early phase of infection and are used for viral DNA synthesis, even in cells that do not actively synthesize cellular DNA. Thymidine is an important substrate of DNA and is supplied in two ways—by the conversion of uridine monophosphate (UMP) to thymidine monophosphate (TMP) via thymidylate synthase (TS) (de novo pathway) and from the systemic circulation by thymidine kinase (TK) (salvage pathway). Acyclovir (ACV) and penciclovir (PCV) are phosphorylated by viral TK and are further phosphorylated to the triphosphate form by cellular enzymes, as shown in the green-shaded boxes. ACV-TP and PCV-TP are incorporated into viral DNA by viral DNA polymerase, which results in chain termination. Foscarnet (PFA) and amenamevir (ASP2151) directly inhibit viral DNA synthesis by inhibiting DNA polymerase and HP, respectively. Nucleotides are used for RNA synthesis in the early phase of viral replication. ACV and PCV are efficiently phosphorylated by viral TK and inhibit viral DNA synthesis because the supply of dGTP is limited in infected cells. This efficient inhibition of viral replication by ACV continues for 5 h after infection. In the late phase of viral replication, ribonucleotides are converted to deoxyribonucleotides by viral RR for viral DNA synthesis as indicated by red arrows. Accordingly, large amounts of dGTP are supplied for DNA synthesis (approximately 60,000 and 90,000 dGTPs per DNA molecule of VZV and HSV, respectively) and this attenuates the inhibition of viral DNA synthesis by reducing the ratio of ACV-TP:dGTP. PFA and ASP2151 directly inhibit viral DNA synthesis and are not influenced by the supply of dGTP, which results in the efficient inhibition of viral growth in contrast to ACV.

**Figure 4 viruses-13-01547-f004:**
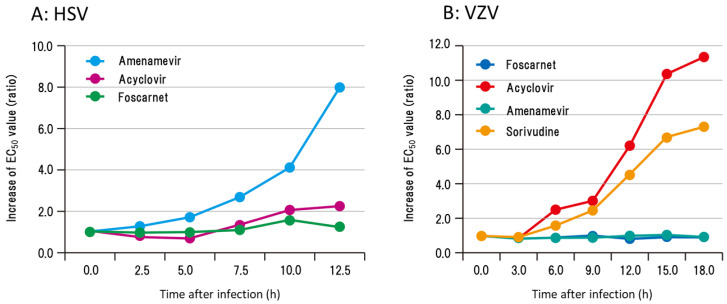
Time course of changes in susceptibility of HSV- and VZV-infected cells to acyclovir, amenamevir, foscarnet, and sorivudine [17,18]. The susceptibility of (**A**) HSV- and (**B**) VZV-infected cells to acyclovir, amenamevir, foscarnet, and sorivudine was assessed serially after infection to examine when and how the susceptibility of infected cells to antiviral drugs might change during the replication cycle of HSV and VZV. EC_50_ values of infected cells to anti-herpetic drugs were determined at the indicated times after infection, and the increase in EC_50_ values is expressed as the ratio of those at 0 h. The authors obtained permission from *Antiviral Research* to reuse these figures [17,18].

**Figure 5 viruses-13-01547-f005:**
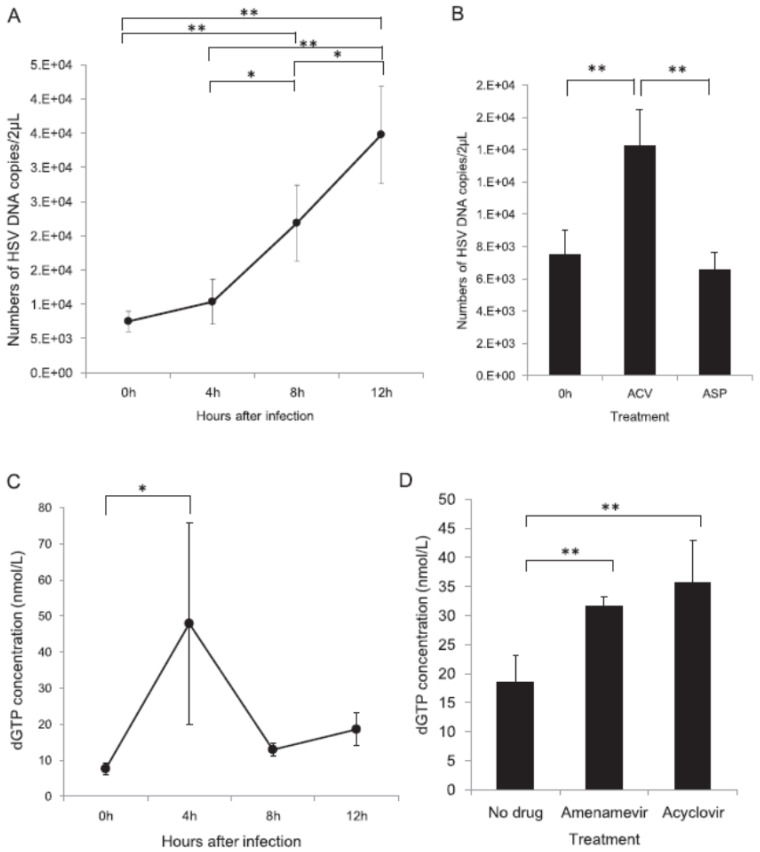
Time-dependent growth of HSV and concentration of dGTP after infection [18]. HSV DNA copy number and dGTP concentration were assessed serially in untreated HSV-infected cells or cells treated with acyclovir and amenamevir. (**A**) Time-dependent increase in HSV DNA copy number after infection. An increase in the copy number was observed later than 4 h after infection. (**B**) Comparison of HSV DNA copy number in cells immediately after infection at 0 h and in cells 12 h after infection treated with 10 times the EC_50_ of amenamevir and acyclovir. The HSV DNA copy number was significantly higher in cells treated with acyclovir than in cells infected at 0 h and cells infected at 12 h and treated with amenamevir. (**C**) Time-dependent changes in the concentration of dGTP after infection. The concentration of dGTP was significantly increased at 4 h after infection and decreased thereafter. (**D**) The concentration of dGTP in infected cells at 12 h without drug treatment was significantly lower than that in infected cells treated with amenamevir and acyclovir. * and ** indicated *p* < 0.05 and *p* < 0.01, respectively. The authors obtained permission from *Antiviral Research* to reuse these figures [18].

**Figure 6 viruses-13-01547-f006:**
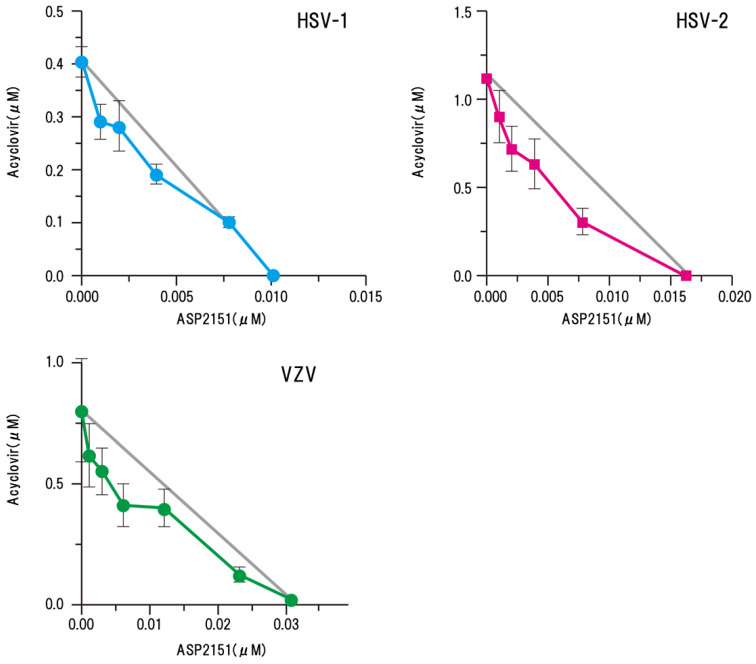
Synergism of amenamevir (ASP2151) with acyclovir against HSV-1, HSV-2, and VZV [15]. Synergism of amenamevir with acyclovir was analyzed by isobologram, and the solid straight line (gray) indicates the theoretical additive antiviral activity in combination with amenamevir and acyclovir. Each point (EC_50_) is the mean ± standard error from four independent experiments. Significant synergism was observed for the combination of amenamevir and acyclovir (*p* = 0.0005), and low concentrations of amenamevir showed stronger synergism with acyclovir than the higher concentrations of amenamevir. The authors obtained permission from *Antiviral Research* to reuse these figures [15].

**Figure 7 viruses-13-01547-f007:**
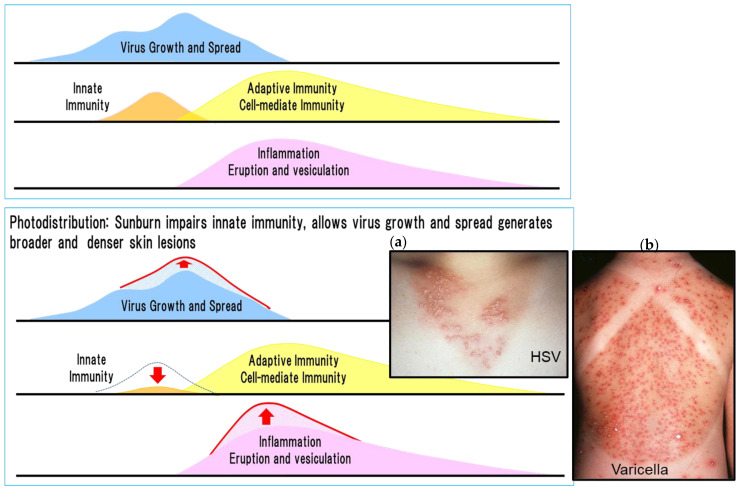
Innate and adaptive immunity and photodistribution [30]. The figure shows the relationship between virus proliferation, innate and adaptive immunity, and rash in HSV and VZV infection. Sunburn modifies virus proliferation, innate and adaptive immunity, and rash as shown by the red arrow in the figure on the right, and causes characteristic photodistribution as shown in the photographs. Exacerbated and dense distribution of skin lesions of HSV and VZV infection related to the inhibition of innate immunity by sunlight exposure (ultraviolet rays). (**a**) A woman in her 20s developed vesicles that were consistently distributed throughout an anterior cervical area exposed to the sun 2 days previously. Denser uniform vesicles and erythema are present in the sun-exposed areas compared with the unexposed areas. The sparse distribution of skin lesions in the area unexposed to sunlight is related to the lack of inhibition of innate immune responses by sun exposure. HSV-1 was present in blister fluid (photodistribution by HSV). (**b**) An 8-year-old girl was diagnosed with varicella with fever and vesicles 3 days after bathing in the sea. Dense uniform vesicles were clustered in the sunburned area, whereas the distribution of eruptions was sparse in the unexposed area where the shoulder straps were located because varicella lesions were inhibited by innate immunity (photodistribution by VZV). In these two cases, innate immunity in the skin was not impaired in areas without sunburn and VZV infection resulted in mild varicella. Thus, innate immunity is important for alleviating dense lesions by inhibiting the growth, spread, and distribution of the virus before inducing adaptive cell-mediated immunity. The photographs were provided by Dr. Yasumoto.

**Figure 8 viruses-13-01547-f008:**
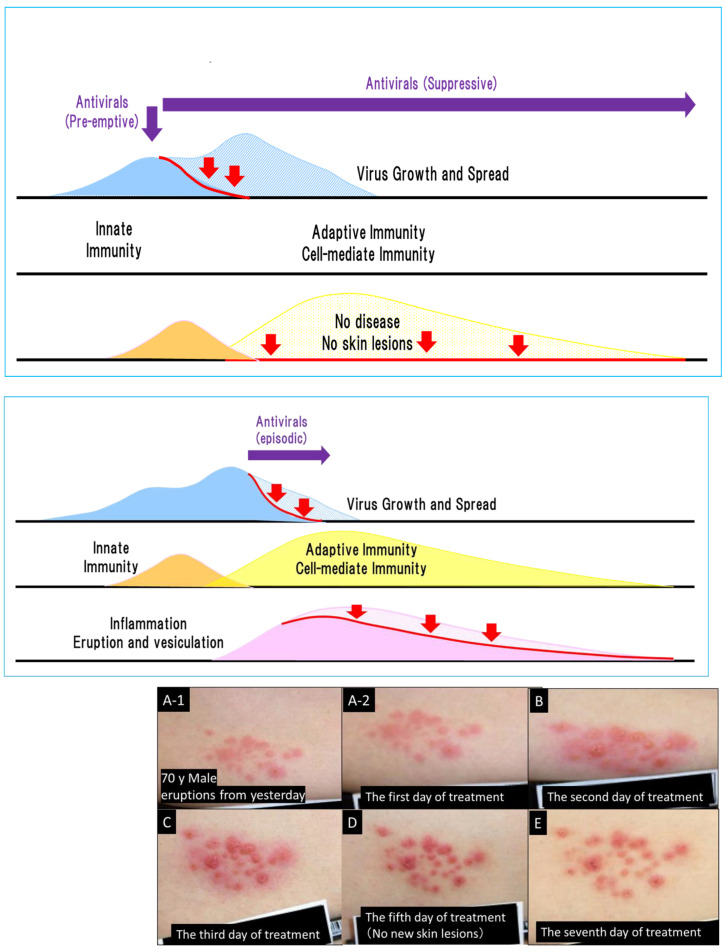
Antiviral treatment at the prodrome and after the onset of herpes zoster. Antiviral therapy inhibits virus growth and spread, as shown by the red arrows. Antiviral treatments started before the prodrome, which are susceptible to photodistribution, include suppressive therapy for HSV, preemptive therapy, and the latter half of the incubation period of varicella. These treatments prevent the onset of overt diseases, which prevents skin lesions. Once the overt disease has occurred, skin lesions continue for 1 week for HSV, 1–2 weeks for varicella, and 3 weeks for herpes zoster. Apparent deterioration of inflammation and improvement of herpes zoster lesions related to cell-mediated immune responses after the initiation of antiviral treatment. Photograph: A 70-year-old male patient noticed a rash on the left-hand side of his waist and treatment with amenamevir was started the day after its appearance [19,22]. Amenamevir inhibited the enlargement of the lesions and new lesion formation and cured the lesions up to the erythema without proceeding to vesicles (abortive infection). (**A-1**,**A-2**) show a panoramic view and a close view of the skin lesions, respectively, on the first day of amenamevir therapy, which was the day after the eruptions were observed. (**B**–**E**) show the 2nd, 3rd, 5th, and 7th days, respectively, of amenamevir treatment. Inflammation in the central part of the eruption increased as assessed by redness and swelling from days 1–5, but the redness and inflammation in the peripheral part of the erythema (red halo) gradually disappeared after day 3. (**B**) shows the skin lesions on the second day after treatment where redness and swelling had increased and spread around the lesions, which indicates that the inflammatory response was augmented by the maturation and enhancement of the cell-mediated immune response to VZV when compared with day 1. (**C**,**D**) show the contrasting course of reduced peripheral inflammation in the red halo areas and increased inflammation in the central part of lesions, with the peak of inflammation at day 5. Urushiol-induced dermatitis peaked on day 3 after antigen contact, and any viral lesions present at the start of treatment appeared as a rash on day 3. There was no new eruption in this patient on day 4, which indicates that antiviral treatment blocked the formation of new lesions and prevented the spread of the eruption and the formation of new lesions. Inflammation represented by redness and swelling was exacerbated despite antiviral treatment. The therapeutic effect of the antiviral drug is not related to the reduction in inflammation, which makes it difficult to determine the therapeutic efficacy of antiviral drugs. The inflammation became more severe between days 3 and 5, even after antiviral treatment. The photograph was provided by Dr. Toyama.

**Table 1 viruses-13-01547-t001:** Amino acid substitutions in the helicase and primase of ASP2151-resistant HSV-1 and HSV-2 mutants, and susceptibility to ASP2151 [14].

Virus	Strains/ Mutants ^a^	Helicase Gene (UL5)	Primase Gene (UL52)	EC_50_ (µmol/L) ^b^	Fold Increase
HSV-1	KOS	- ^c^	-	0.037	-
	K2151^r^m	G352V, M355I	S364G, R367X ^d^	131.8	3562
	K2151^r^m#B9	G352V, M355I	S364G, R367H	105.4	2849
	K2151^r^m#D9	G352V, M355I	S364G	19.6	530
	K2151^r^m#G11	G352V, M355I	S364G	28.2	762
	K2151^r^m#H10	G352V, M355I	S364G, R367H	118.0	3189
HSV-2	Lyon	-	-	0.12	-
	L2151^r^m#C1	K355N, K451R	-	>150	>1250

^a^ K2151rm#B9, K2151rm#D9, K2151rm#G11, and K2151rm#H10 were derived from K2151rm and L2151rm8#C1 was derived from L2151rm by single plaque isolation. ^b^ The 50% effective concentration (EC50) was calculated via nonlinear regression analysis using a sigmoid-Emax model from one (HSV-2) or three (HSV-1) independent experiments performed in triplicate. ^c^ Identical to the parental sequence, or no substitutions were observed. ^d^ ‘X’ indicates Arg (R) or His (H) due to the detection of a mixed-base signal at the 367th Arg codon. The authors obtained permission from *Biochemical Pharmacology* to reuse this table [14].

**Table 2 viruses-13-01547-t002:** EC_50_ values of ASP2151 and ACV against ACV-resistant or ACV-susceptible HSV-1, HSV-2, and VZV strains (plaque reduction assay) [15].

Virus	Strains	EC50 (95% Confidence Interval) (µM) ^a^		Susceptibility (Amenamevir/Acyclovir) ^d^
		Amenamevir (ASP2151)	Acyclovir	
HSV-1	KOS	0.010 (0.0082–0.012)	0.400 (0.32–0.50)	+/+
	A4-3	0.067 (0.049–0.091)	1.15 (98.8–133)	+/−
HSV-2	Genital isolate	0.012 (0.006–0.023)	1.34 (0.51–3.56)	+/+
	Whitlow 2	0.012 (0.006–0.022)	65.9 (31.9–136)	+/−
VZV	Kawaguchi ^b^	0.064 (0.043–0.094)	1.61 (0.99–2.63)	+/+
	TK-deficient mutant	0.068 (0.052–0.088)	12.8 (9.5–17.3)	+/−
	A2 ^c^	0.11 (0.078–0.16)	11.5 (6.5–20.3)	+/−
	A3 ^c^	0.11 (0.049–0.26)	19.2 (11.1–33.1)	+/−
	A7 ^c^	0.065 (0.045–0.093)	41.4 (21.6–79.2)	+/−
	A8 ^c^	0.10 (0.062–0.162)	82.2 (72.7–92.9)	+/−

^a^ Means of four independent experiments. ^b^ Parental strain of TK-deficient mutants, A2, A3, A7, and A8. ^c^ DNA polymerase mutant. ^d^ Susceptibility of virus strains to each compound: +, susceptible; -, resistant. The authors obtained permission from *Antiviral Research* to reuse this table [15].

**Table 3 viruses-13-01547-t003:** Cumulative number of adverse drug reactions related to oral anti-herpetic drugs (as of 16 May 2021).

System Organ Class	2015	2016	2017	2018	2019	2020
Nervous system disorders	221	257	272 (15)	255 (21)	271 (14)	254 (5)
Renal and urinary disorders	235	212	144 (1)	195 (5)	211 (6)	134 (6)
Psychiatric disorders	31	31	29 (1)	10 (1)	22 (1)	15 (0)
Infections and infestations	11	9	21 (5)	38 (9)	40 (1)	16 (2)
Skin and subcutaneous tissue disorders	14	6	27 (7)	25 (15)	34 (10)	22 (10)
General disorders and administration site conditions	17	15	19 (4)	36 (19)	24 (7)	16 (3)
Metabolism and nutrition disorders	7	12	13 (2)	33 (11)	16 (2)	23 (4)
Others	69	78	105 (11)	108 (25)	98 (11)	82 (12)
Total	605	620	630 (44)	700 (106)	716 (52)	562 (42)

Number (including amenamevir). Adverse drug reaction reports to the PMDA are reported by pharmaceutical companies or medical institutions, and the PMDA has not individually evaluated their relevance to the pharmaceutical products. In addition, because the number of reported adverse drug reactions varies depending on the number of patients administered each product and the availability of information from pharmaceutical companies, it is not possible to evaluate or compare the safety of pharmaceutical products based on the number of reported adverse drug reactions. (Japanese Adverse Drug Event Report database: https://www.info.pmda.go.jp/fsearchnew/jsp/menu_fukusayou_base.jsp, accessed on 30 July 2021).

## Data Availability

Not applicable.

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
