# Peer review of "Amenamevir, a Helicase-Primase Inhibitor, for the Optimal Treatment of Herpes Zoster"

_viruses, 2021, doi:10.3390/v13081547_

Round 1

Reviewer 1 Report

This is an interesting article, however there is mild repetition and I would recommend considerinng the order in which information is presented, although the effect on the reader is quite minor and doesn't degrade comprehension. It just requires more attention in places to extract the intended meaning. Suggested edits are given below.

Line numbers refer to the reviewed pdf:

Line 21 - Please insert a comma rather than full-stop after "Japan", thus no new sentence after "Japan" and continuation of sentence via lower case "s"; i.e. "Japan, side effects [...]".

Line 58 - "form" is a typographic error and should read: "fork".

Between lines 155 and 156 - Top row of Table 1 - While acknowledging that the table is reused with permission: Suggest moving the word "mutants" onto the next line rather than breaking onto it at "mu-", similarly "increase to the next line rather than breaking onto it with "in-".

Line 211 - There is a legend reference to Figure 3 to part "(a)" and also part "(b)" on line 214, which don't appear on the figure itself. On line 211, the sentence beginning "The early phase of viral replication and [...]" would make more sense if it were "In the early phase of viral replication [...]"

Line 214 - There is reference to a circle in Figure 3, but there is no circle? Does this refer to one of the green-shaded boxes?

Line 220 - There is reference to a circle in Figure 3, but there is no circle? Does this refer to one of the green-shaded boxes?

Line 352 - Would be more correct to change "since replication fork is" to "since replication fork formation is"

Author Response

Thank you for reviewing our manuscript. We have revised the manuscript according to your suggestions.

Line 21 - Please insert a comma rather than full-stop after "Japan", thus no new sentence after "Japan" and continuation of sentence via lower case "s"; i.e. "Japan, side effects [...]".

Response

As suggested by the reviewer, the sentence has been rephrased as underlined. “Post-marketing surveillance of amenamevir in Japan, side effects with significant potential risk identified by the Japanese Risk Management Plan were thrombocytopenia, gingival bleeding, and palpitations, although none of these were serious.

Line 58 - "form" is a typographic error and should read: "fork".

Response

As suggested by the reviewer, "form” has been corrected "fork"

Between lines 155 and 156 - Top row of Table 1 – While acknowledging that the table is reused with permission: Suggest moving the word "mutants" onto the next line rather than breaking onto it at "mu-", similarly "increase to the next line rather than breaking onto it with "in-".

Response

As suggested by the reviewer, the hyphenated words have been changed to "mutants" and "increase" without hyphens by changing the line.  

Line 211 - There is a legend reference to Figure 3 to part "(a)" and also part "(b)" on line 214, which don't appear on the figure itself. On line 211, the sentence beginning "The early phase of viral replication and [...]" would make more sense if it were "In the early phase of viral replication [...]"

Line 214 - There is reference to a circle in Figure 3, but there is no circle? Does this refer to one of the green-shaded boxes?

Line 220 - There is reference to a circle in Figure 3, but there is no circle? Does this refer to one of the green-shaded boxes?

Response

As addressed by the reviewer, Figure 3 legend needs improvement.

  1. (a) and (b) have been eliminated.
  2. “as marked by a circle” has been eliminated.
  3. Underlined parts have been changed to improve the paragraph.

Acyclovir (ACV) and penciclovir (PCV) are phosphorylated by viral TK and are further phosphorylated to the triphosphate form by cellular enzymes as shown in the green-shaded boxes.

In the early phase of viral replication nucleotides are used for RNA synthesis. ACV and PCV are efficiently phosphorylated by viral TK and inhibit viral DNA synthesis because the supply of dGTP is limited in infected cells. This efficient inhibition of viral replication by ACV continues for 5 h after infection. In the late phase of viral replication ribonucleotides are converted to deoxyribonucleotides by viral RR for viral DNA synthesis as indicated by red arrows. Accordingly, large amounts of dGTP are supplied for DNA synthesis (approximately 60,000 and 90,000 dGTPs per DNA molecule of VZV and HSV, respectively)

  1. Explanation of red arrows and * asterisks is added to clarify the pathway from rGDP to dGTP in the figure. “RR increases dGTP in the late phase of infection and the ratio of ACV-TP or PCV-TP per dGTP for DNA synthesis decreases, resulting in weakening the action of ACV or PCV. HPI is not affected by the increase of dGTP (Fig. 4).

Line 352 - Would be more correct to change "since replication fork is" to "since replication fork formation is"

Response

As suggested by the reviewer, "since replication fork is" has been changed to "since replication fork formation is"

Reviewer 2 Report

This review introduced amenamevir as a helicase-primase inhibitor (HPIs) with novel mechanisms of anti-herpetic action and compared its antiviral features with acyclovir. The title and abstract are appropriate for the content of the text. Also, the manuscript is well-organized and documented. However, two points need to be revised. Firstly, in Figure 1, two words (LeadingX, LaggingX) are mislabeled. Secondly, Figure 5 is quite blurry. Publication quality images should be used with 300dpi or above.

Overal, I recommend publication of this review after minor revisions. The review highlights the therapeatic potential of amenamevir as HPIs in anti-herpes treatment. It will be of interest to researchers interested in exploring novel antiviral compounds particularly with anti-herpetic activity.

Author Response

Thank you for reviewing our manuscript. We have revised the manuscript according to your suggestions.

Firstly, in Figure 1, two words(LeadingX, LaggingX) are mislabeled.

Labeling of “Leading strand” and “Lagging stran”

Response

As addressed by the reviewer, The locations pointed to by “Leading strand” and “Lagging strand” are misleading, so I have clearly shown them in the new Fig. 1 so that they are not misunderstood.

Secondly, Figure 5 is quite blurry. Publication quality images should be used with 300dpi or above.

Response

As addressed by the reviewer, quality of Fig. 5 should be improved. Our original figure would be sufficient and we will ask publisher to use the original figure with high quality.